# Empirical evidence on the impact of the "new round" of Sino-US trade frictions on China's foreign trade industrial policy and high-quality development

He Leihua[1], Sun Fan●[2]*

1 School of Economics & Management, Northwest University, Xi'an, Shaanxi Province, China, 2 School of Management and Economics, North China University of Water Resources and Electric Power, Zhengzhou, Henan Province, China

* m13244623829@163.com

**Data Availability Statement:** The data used in the paper are available in public databases,the data in the article come from the Statistical Yearbook on

## Abstract

The global attention on industrial policy's effectiveness spans across various sectors, particularly in international trade where the focus is on enhancing the quality of foreign trade, which is crucial to understand how research hotspots and key issues can synergize. We consider the 2018 "new round" of Sino-US trade friction as an external event and analyze panel data from 2009 to 2022 comprising 1141 Chinese A-share listed enterprises. Our empirical research unequivocally demonstrates that industrial policy has significantly propelled China's foreign trade towards high-quality development by 1.8240%. We conducted rigorous tests for robustness, heterogeneity, and endogeneity. Additionally, our results reveal that inhibitory influences arising from industrial policy on total assets or R&D investment attenuate this promotion effect; There exists a diminishing promotion effect between total assets and R&D investment. Our paper provides valuable insights into understanding their logical relationship while offering guidance for enterprises, markets, and governments in overcoming challenges collectively.

## 1. Introduction

China's "14th Five-Year Plan for the Development of High-quality Foreign Trade" clearly states that by 2035, there will be a remarkable advancement towards high-quality development in foreign trade. This includes optimizing the structure of trade, achieving a more balanced import-export ratio, significantly enhancing innovation capacity, making positive progress in green and low-carbon transformation, improving security guarantee capabilities, and gaining new advantages in international economic cooperation and competition [1]. Throughout the past forty years, China has actively participated in global value chains and engaged in the international division of labor by leveraging its labor force and natural resources [2]. However, recent years have seen Western nations led by the United States adopting economic strategies such as "decoupling" and "de-risking," driven by anti-globalization sentiments. These policies

the official website of the China Bureau of Statistics at https://www.stats.gov.cn/sj/ndsj/.

**Funding:** Fund project: This research was supported by the project "Industrial Logic of High-quality Development of China's Foreign Trade under New Trade Protection" (L.H.; 20FJLB008) of the National Social Science Foundation and University-level project "Research on the Trade Protection Effect of China's Industrial Policy under Market Forces" (F.S.; NCWUYC-202315041). Author Contributions (The funders are also the authors of this article, so their contributions are consistent, namely as follows.) Conceptualization: He Leihua, Sun Fan. Formal analysis: Sun Fan. Investigation: He Leihua. Methodology: He Leihua, Sun Fan. Writing – original draft: He Leihua, Sun Fan. Writing – review & editing: He Leihua, Sun Fan.

**Competing interests:** The authors have declared that no competing interests exist.

have undermined political trust between China and the United States while further intensifying strained economic and trade relations. Consequently, they have also introduced significant uncertainties to the global economy.

Industrial policy is a government-developed and implemented policy framework aimed at promoting the growth of specific industries or economic sectors within a country or region, enhancing their competitiveness, increasing productivity, generating employment opportunities, and achieving objectives of economic growth and development [3, 4]. However, industrial policy can be subject to controversy due to potential issues arising from government intervention in the market such as unfair competition, uneven resource allocation, and challenges in policy implementation [5, 6]. Therefore, this study investigates the effectiveness of industrial policy for facilitating high-quality development in China's foreign trade by examining its efficacy under the backdrop of the recent Sino-US trade friction in 2018. Additionally, it explores when industrial policy proves effective and suggests strategies for maximizing its effectiveness.

The existing literature has given some answers to the appeal question. First of all, how to scientifically and quantitatively evaluate the index of industrial policy, the existing research mainly includes the following three aspects: First, industrial policy is processed in the way of dummy variables, that is, after a policy is assigned a value of 1 in the current year and after that, and after a policy is assigned a value of 0 before that, the relevant model is analyzed [7–9]. However, this measurement method makes the quantitative evaluation of the specific effects of industrial policy extremely rough, and cannot clearly answer the question of when and how industrial policy is effective; Second, use Python software to capture the text of relevant documents such as enterprise annual reports and five-year plans, and then construct industrial policy indicators according to relevant word frequency [10–12], but this measurement method is extremely strict in the selection of related words, and the description of industrial policy by using texts such as "five-year plan" reflects more prior measurement; Third, selective industrial policies are also described in terms of financial subsidies and tax incentives issued by the government to enterprises [13–15], considering that it is necessary to quantitatively analyze the impact of industrial policy and its implementation under the new round of Sino-US trade friction, this paper continues to use the last method to characterize industrial policy indicators based on government macro-level measures such as taxation and subsidies [16], which makes the paper more accurate in the use of data. At the same time, it also shows the characteristics of the combination of macro government and micro enterprises.

Secondly, within the context of recent Sino-US trade friction, various methods have been employed to showcase the high-quality development of foreign trade. One method involves analyzing the import and export ratio as discussed by Zhang Yan and Fu Xin (2022) and Jidong et al.,9 (2022) [17, 18]. Another perspective considers total factor productivity (TFP) as an indicator for representing high-quality development in foreign trade, highlighted in studies by Song Min et al., (2021) and Huan (2021) [19, 20]. A third approach involves assessing word frequency related to innovation, coordination, green practices, openness, and sharing. This method proposed by Alschner et al., (2018) and Zhao Jingmei et al., (2023) [21, 22], aims to capture the essence of high-quality development in foreign trade. Given the maturity of TFP method in constructing indicators for high-quality development in foreign trade which enables a more nuanced reflection of enterprise data at micro level, this paper opts for OP method based on TFP to construct these indicators. Additionally, to ensure robustness analysis, the text mining method applied to enterprise annual reports is employed to construct supplementary indicators.This dual-method approach enhances reliability and comprehensive understanding towards high quality development from both macro-level perspective as well as micro-level perspective.

Moreover, from the perspective of theoretical research, With the deepening of economic globalization and the trade friction between China and the US, As the largest developing country in the world, China, The influence effect of the issued industrial policy on the development of foreign trade has gradually become the focus of attention, Scholars mainly analyze the impact of industrial policy on the high-quality development of foreign trade from the models of local equilibrium, general equilibrium and evolutionary game, For example, Tan Linyuan and Li Xiande (2022) by building a local equilibrium model, including multiple agricultural industries [23]. To simulate and predict the impact of the increase on the demand and supply of domestic barley industry. The results indicate an increase in import duties on barley, Domestic barley prices and planting area have increased, Protecting the domestic barley industry, Reducing international exports; Shikar et al., (2021) constructed a general equilibrium model in order to study the relationship between industrial policy and the economic development in the Oromia region [24]. Anticipate the impact of industrial policy, And research shows that industrial policy will have the biggest impact on manufactured goods exports. The continent is trying to create jobs through structural transformation catch-up. To realize the economic development of the Oromia region; Yan Zhenkun and Liu Yifang (2023) Based on the evolutionary game model between the central government, the pioneering local government and the follow-up local government [25]. It is concluded that the tax rate policy of the central government, the investment in local competitive industries, the degree of opening to the outside world, the degree of support of the pioneering local government to the local emerging industries and the understanding of the local government in the region are important factors affecting the consistency of the strategic goals of governments at all levels. However, these models do not clearly describe the mechanism of industrial policy affecting the high-quality development of foreign trade, but simply explain the effect of industrial policy.

Finally, Empirical research has established the significant impact of industrial policy on promoting high-quality development in foreign trade, with scholars recognizing its scale and innovation. For instance, Xiaolin (2021) found that increased government subsidies to energy enterprises in China lead to higher fixed assets and R&D investments but may reduce the efficiency of OFDI [26]. Jiang Yun and Li Wei (2023) discovered that intensifying agricultural subsidy policies directly increases exporters' income and indirectly raises importers' income [27]. Wang et al., (2020) demonstrated that augmenting subsidies or tax incentives for both state-owned and non-state-owned environmental protection enterprises by the government contributes to technological innovation and enhances enterprise-level foreign trade development [28]. Zhang Wenqing et al., (2023) revealed that increasing government subsidies stimulates green technology innovation within enterprises while promoting their foreign trade development [29]. However, further clarification is required regarding the specific effects of these two aspects of industrial policy on achieving high-quality development in foreign trade, as well as determining which measures are more effective for enhancing foreign trade development. This will provide better insights into when and how industrial policy can be effectively implemented.

Based on the potential marginal contributions of this paper, three key points can be outlined as follows: Firstly, an examination is conducted on the effectiveness of macro-level industrial policies by utilizing micro-level data from 1141 A-share listed enterprises spanning from 2009 to 2022. This approach enhances the credibility of the research findings since it relies on a robust dataset consisting of A-share listed entities. Secondly, a more comprehensive perspective on the impact of industrial policy on high-quality foreign trade development is provided by extending the analysis to encompass various types of enterprises. By acknowledging and considering variations across different enterprise types, this inclusive approach contributes to a nuanced understanding of the subject matter. Thirdly, in order to deduce how industrial

policy influences China's foreign trade development towards higher quality, a theoretical analysis mechanism is employed. This theoretical foundation serves as a reliable framework for subsequent empirical analysis and interpretation validation of observed outcomes.

## 2. "New round" of Sino-US trade friction

### 2.1. Origin of friction

The year 2018 witnessed a transformative juncture in Sino-US trade relations, marked by a sequence of consequential events [30]. In the beginning of 2018, President Donald Trump initiated tariffs on steel and aluminum products, with a rate of 25 percent for steel and 10 percent for aluminum, specifically focusing on imported goods. This measure not only evoked vehement opposition from certain allies but also laid the groundwork for the subsequent evolution of Sino-US trade disputes. Subsequently, in March, the United States initiated a Section 301 investigation against China, alleging unfair trade practices encompassing forced technology transfer and intellectual property theft. Reactive to this, the United States proclaimed 25 percent tariffs on approximately $50 billion worth of Chinese goods, with a focus on high-tech and industrial manufacturing products.The escalation of trade tensions reached a zenith in July when China retaliated with tariffs on US goods, intensifying a trade war characterized by the gradual imposition of tariffs and other trade restrictions by both sides. Despite reaching a phase one trade agreement in late 2018, its implementation did not commence until January 2019. Under the terms of the agreement, China committed to augmenting its purchases of U.S. goods, enhancing the protection of intellectual property rights, and undertaking specific market access commitments. In response, the United States deferred the implementation of certain planned tariffs initially scheduled for December 2019. This sequence of events not only altered the dynamics of trade relations between China and the United States but also exerted a profound influence on the global economy.

### 2.2. Cause of friction

The aforementioned issue persisted over an extended duration, primarily evident in the trade disparity.The United States has long been concerned about the trade deficit with China. Although China has long been one of the United States' most important trading partners, the United States views the trade relationship as unfair, especially since China exports far more to the United States than the United States exports to China. This has led the US government to try to reduce the trade deficit through trade instruments; Second, intellectual property issues loom large as another central factor. The United States claims that China takes unfair measures in technology transfer and intellectual property rights, accusing China of obtaining technology through forced technology transfer, patent infringement and theft of trade secrets, thereby harming the interests of American companies. This issue has become the core of the trade war dispute, involving concerns about the technological and innovative competitiveness of both sides; Moreover, the market access issue highlights dissatisfaction with China's barriers to market access for foreign companies. The United States has complained that China uses restrictive regulations and administrative measures that impede market access for foreign companies. The United States urges China to open more sectors and reduce restrictions on foreign investment in order to promote fairer market access. Finally, industrial policy issues have attracted attention, particularly U.S. concerns about China's "Made in China 2025" plan. The plan aims to upgrade Chinese industries from low to high value-added, but the United States is concerned that it could distort the market through state subsidies and other means, giving Chinese companies an unfair competitive advantage. The administration also highlighted a range of

issues related to economic security, including control of critical technologies and investment in sensitive areas, reflecting concerns about technology leaks and national security threats.

## 2.3. Friction effect

Within a relatively concise timeframe, China has achieved the industrialization process that took the West two centuries to complete, and the rise and development of small, medium and micro enterprises have contributed significantly to this success. However, it is worth noting that in 2018, China's exports to the United States were subjected to high tariffs, resulting in some enterprises being forced to relocate production capacity to other countries in order to reduce export costs. In this context, some multinational companies and countries began to actively advocate the localization, diversification and inshore of the industrial chain and supply chain, in order to effectively deal with the impact of unpredictable extreme events on the industry, to ensure the normal operation of enterprises and promote the steady development of the industry. In recent years, Chinese enterprises have faced serious business challenges, especially small and medium-sized enterprises. According to data from the 2022 China Enterprise Innovation and Entrepreneurship Survey (ESIEC), a survey of newly registered industrial and commercial enterprises over the past decade shows that only 35 percent can be confirmed to be operating, while 57.3 percent cannot determine their business status. Since 2014, the number of business cancellations has continued to increase, with 5.1% of small enterprises cancelling in 2022, while the proportion of micro-enterprises cancelling is even higher, reaching 10.8% [31]. This phenomenon highlights the serious challenges faced by Msmes in the current environment, with a particular impact on the Chinese economy.

Chinese enterprises face significant challenges in expanding their global market presence due to high tariff barriers, numerous targeted legal and administrative regulations, market rules, and other obstacles. These factors considerably amplify the explicit and implicit costs associated with international expansion, thereby hindering overseas market growth and substantially reducing product profitability.This effect is manifested in the decline of enterprise performance, the weakening of competitiveness, the weakening of market power, the reduction of control power, especially the reduction of large enterprises in the market share. A large number of Chinese foreign trade enterprises are facing serious challenges, including but not limited to the reduction of orders, excess capacity, and may even face the potential risk of closure and bankruptcy. The array of challenges outlined above underscores the significant influence wielded by trade barriers and market regulations on the operational endeavors of enterprises, thereby presenting a formidable obstacle to the enduring progress of China's foreign trade enterprises. In light of these circumstances, this paper strategically designates the year 2018 as the temporal focal point of the event's impact. The objective is to conduct a comprehensive investigation into the efficacy, timing, and strategies through which industrial policies can effectively facilitate China's international trade advancement.

## 3. Theoretical analysis framework

Suppose that there is an enterprise W with foreign trade business in the economy, the output of the normal production of foreign trade product X is $Q$, and the reduced output of product X caused by Sino-US trade friction is $Q_1$. Suppose that enterprise W satisfies the characteristics of the form of production function and the return to scale unchanged of Corduggas, then $Q_1$ increases with the increase of $Q$.

With reference to the model setting of Guo Jie et al., (2019) [32], assuming that two production factors $K$ and $L$ are input, the output of enterprise W is $F(K, L)$. The proportion of output reduction caused by Sino-US trade friction in total output is $u$, where $u \in [0,1]$, when $u = 0$, it

means that enterprise W is not affected by Sino-US trade friction; When $u = 1$, it means that the foreign trade product X of enterprise W stops production and exits the market after being affected by Sino-US trade friction. At this time, the actual output $Y$ of foreign trade product X is actually $(1-u) F (K, L)$, as shown in Eq (1) below:

$$Y = (1 - u)F(K,\ L) \tag{1}$$

The expression of output $Q_1$ of product X lost due to trade friction is as follows (2):

$$Q_1 = \lambda(u)F(K,\ L) \tag{2}$$

Considering that the losses caused by trade frictions decrease in proportion to trade inputs, $\lambda(u)$ can be expressed as follows:

$$\lambda(u) = \frac{(1 - u)^{\frac{1}{a}}}{A} \tag{3}$$

Where $A$ represents the level of technological innovation to produce product X, and the higher the level, the less losses caused by trade frictions. In addition, it can be seen from Eq (3) that the loss caused by trade friction is a monotonically decreasing "concave" graph function with the first derivative less than or equal to 0 and the second derivative greater than 0. Substituting Eq (3) into Eq (2) then $Y_1$ can be expressed as:

$$Q_1 = \frac{(1 - u)^{\frac{1}{a}}}{A}F(K,\ L) \tag{4}$$

At the same time, after sorting out Formula (4), it can be obtained as follows:

$$1 - u = \frac{(AQ_1)^a}{F^a(K,\ L)} \tag{5}$$

Then substitute Eq (5) into Eq (1) to finally get the actual output of enterprise W's foreign trade product X as shown in Eq (6):

$$Y = (AQ_1)^a F^{1-a}(K,\ L) \tag{6}$$

From Formula (6), it can be seen that product X is not only affected by the potential output F, including factors K and L, but also affected by the level of technological progress $A$ and the loss $Q_1$ caused by trade friction.

From the perspective of producer decision-making, in order to minimize the cost of foreign trade product X, given the cost loss $Q_1$ and unit potential output $F$ caused by trade friction, the minimum production cost per unit $(AQ_1)$ is $c_1$, and the minimum production cost per unit potential output $F$ is $c_2$, that is, there is a functional relationship as shown in Eq (7) below:

$$\begin{cases} min\ \ C(c_1, c_2) = c_1(AQ_1) + c_2F \\ s.t.\ \ (AQ_1)^a F^{1-a} = 1 \end{cases} \tag{7}$$

In order to relate enterprise production to cost, we can construct a Lagrange function as shown in Eq (8), where $m$ is a Lagrange multiplier and is not equal to 0:

$$L = c_1(AQ_1) + c_2F + m[1 - (AQ_1)^a F^{1-a}] \tag{8}$$

Derivation of $AY_1$ and $f$ can be obtained as follows (9):

$$\begin{cases} \dfrac{\partial L}{\partial(AQ_1)} = c_1 - ma(AQ_1)^{a-1}F^{1-a} \\ \dfrac{\partial L}{\partial F} = c_2 - m(1-a)(AQ_1)^{a}F^{-a} \end{cases} \tag{9}$$

After sorting, the first-order optimal solution can be obtained, as shown in Eq (10) below:

$$\frac{(1-a)AQ_1}{aF} = \frac{c_2}{c_1} \tag{10}$$

Assuming that the price of product X is $P_1$, the total revenue $TR$, total cost $TC$ and total profit $TP$ of enterprise W are shown in Eq (11) below:

$$\begin{cases} TR = P_1 Q \\ TC = c_1(AQ_1) + c_2 F \\ TP = TR - TC \end{cases} \tag{11}$$

If firm W is in a long-run equilibrium in a perfectly competitive market, then its total profit $TP$ is 0, then:

$$P_1 Q = c_1(AQ_1) + c_2 F \tag{12}$$

After sorting out Eqs (10) and (12) and assuming that the price of this part of the product lost is $P_2$, we can get:

$$Q_1 = \frac{aP_1 Q}{c_1 A} = \frac{a}{c_1 A} P_1 Q = \frac{a}{c_1 A} \cdot G \tag{13}$$

In Formula (13), it can be found that A represents the level of production technology progress, that is, the technical effect $A$, $P_1Q$ represents the total revenue of product X, that is, the scale effect $G$. Then take the logarithm of Eq (13) to get Eq (14):

$$LnQ_1 = Lna - Lnc_1 + LnG - LnA \tag{14}$$

At the same time, industrial Policy is introduced into the production function as an input factor, that is, $c_1 = c_1(Policy)$, $G = G(Policy)$ and $A = A(Policy)$, then Eq (14) can be expressed as follows:

$$LnQ_1 = Lna - Lnc_1(Policy) + LnG(Policy) - LnA(Policy) \tag{15}$$

The growth in the production of internationally traded goods is a crucial indication of China's achievement of high-quality development in its foreign trade sector. Therefore, this study suggests that the development of industrial policies has an impact on the high-quality advancement of foreign trade, encompassing both the benefits derived from economies of scale and technological advancements as illustrated in Fig 1 below, where M represents either scale advantages or technical effects or both. However, further empirical investigation is necessary to explore the extent, directionality, and statistical significance of this influence.

## 4. Benchmark models, data sources, and variable measures

### 4.1. Reference model

In an effort to examine the influence of industrial policies on China's progress in high-quality foreign trade, this study centers its attention on the trade dispute that occurred between China

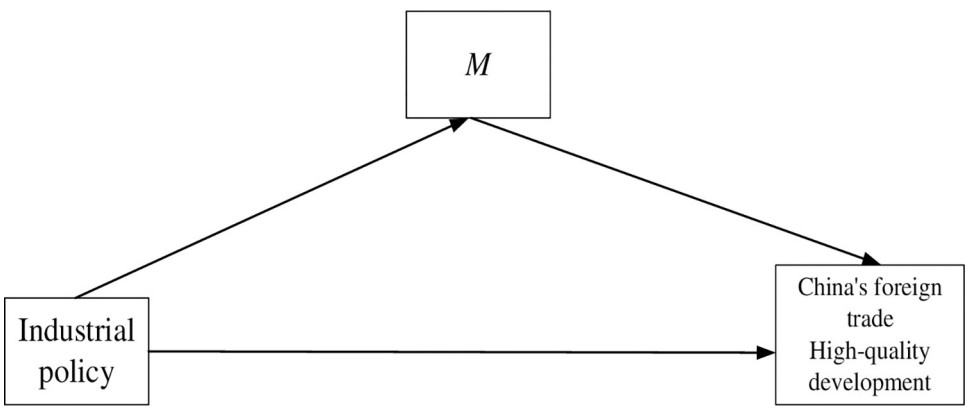

**Fig 1. Action mechanism diagram.**

and the United States in 2018. To facilitate analysis, we have opted to explore the distinctive model outlined in Eq (16) provided below.

$$Fthd_{i,t} = a_0 + a_1 Policy_i \times Treat_t + a_2 X_{i,t} + \alpha_i + \beta_t + \varepsilon_{i,t} \tag{16}$$

Among them, $Fthd_{i,t}$ represents the high-quality foreign trade development level of enterprise $i$ in year $t$, $a_0$ is the intercept term, $Policy_i$ represents the industrial policy level of enterprise $i$ (mainly represented by financial subsidies $Fs$, tax incentives $Tp$ and the integrated $Ip$ of the two), $Treat_t$ represents the virtual variable of time, which is 1 in 2018 and thereafter, otherwise 0. Using the interaction term of $Policy_i$ and $Treat_t$ as the core independent variable, the value, shape and significance of coefficient $a_1$ can reflect the impact of industrial policy on the $Fthd$. $X_{i,t}$ represents the control variable of enterprise $i$ for $t$ years. $\alpha_i$ and $\beta_t$ represent firm fixed effects and time fixed effects, respectively, $\varepsilon_{i,t}$ represent random disturbance terms.

In order to further study how industrial policy affects the $Fthd$, this paper adopts the step-by-step regression equation shown in Eq (17) and Eq (18) as follows:

$$Z_{i,t} = b_0 + b_1 Policy_i \times Treat_t + b_2 X_{i,t} + \alpha_i + \beta_t + \varepsilon_{i,t} \tag{17}$$

$$Fthd_{i,t} = c_0 + c_1 Policy_i \times Treat_t + Z_{i,t} + c_2 X_{i,t} + \alpha_i + \beta_t + \varepsilon_{i,t} \tag{18}$$

Where,$Z_{i,t}$ represent the scale effect and technological effect of enterprise $i$ in $t$ years, and the value, positive and negative shape and significance of coefficient $b_1$ can reflect the impact of industrial policy on enterprise scale or enterprise technology. The size of coefficient $c_1$, and the utilization of coefficient a1 enables the comparison and determination of whether industrial policy affects the high-quality development of foreign trade by influencing enterprise scale or technology.If the sobel test passes and the coefficient $c_1$ is less than the coefficient $a_1$, the above mechanism of action is proved.

## 4.2. Data source

In this paper, in 2009 to 2022 China a-share listed companies as the research object (before 2008 and 2023 after data missing, cannot provide reliable experimental results), core data from the Tai'an (CSMR) database, Wind database and China city statistical yearbook, etc., to cross check data and 1% tail processing, and select the research period has been normal operation of the listed companies, finally screening meet the conditions of 1048 China a-share listed companies. At the same time, according to the national economic development, it is divided into four regions: eastern, central, western and northeast. According to the classification standards

of the securities Association in 2012, enterprises are divided into the primary industry, the secondary industry and the tertiary industry (The primary industry begins with the industry code A, and the secondary industry begins with the industry code B, C, D, E, and F, and those other than the primary industry and the secondary industry are the tertiary industry).

### 4.3. Measures of correlated variables

**4.3.1. Measurement of industrial policy.** Common industrial policy indicators of fiscal subsidies ($Fs$) and tax incentives ($Tp$), this paper choose the CSMR database, with the government to the enterprise of the logarithm of production subsidies to fiscal subsidies, with the government returned to the enterprise divided by the taxes and fees and received the sum of the taxes and fees to express tax incentives, and enterprises are usually fiscal subsidies and tax incentives in parallel. Therefore, this paper first uses the coefficient of variation method of Eq (19) to determine the weights of fiscal subsidies and tax incentives [33, 34], avoiding the error of artificial weight setting.

$$\begin{cases} Q_{Fs} = \dfrac{Sd_{Fs}/Av_{Fs}}{Sd_{Fs}/Av_{Fs} + Sd_{Tp}/Av_{Tp}} \\ Q_{Tp} = \dfrac{Sd_{Tp}/Av_{Tp}}{Sd_{Fs}/Av_{Fs} + Sd_{Tp}/Av_{Tp}} \end{cases} \tag{19}$$

Where $Q_{Fs}$ represents the weight of financial subsidies, and the calculated size is 0.6889. The quotient of standard deviation $Sd_{Fs}$ and average $Av_{Fs}$ of financial subsidies in the sample is divided by the sum of financial subsidies and tax incentives after the same practice, and the weight $Q_{Tp}$ of tax incentives is calculated to be 0.3311. Then, the dimensionless treatment of fiscal subsidies and tax incentives is carried out in Formula (20):

$$\begin{cases} Fs_n = Q_{Fs} \dfrac{A_n - m}{M - m} \\ Tp_n = Q_{Tp} \dfrac{A_n - m}{M - m} \end{cases} \tag{20}$$

Among them, in order to avoid the unrobustness of the results caused by policy differences among different industries, this paper calculates the industrial policy level of each enterprise by three major industries. $A_n$ represents the actual value of financial subsidies or tax incentives in the NTH data of the three major industries, and $M_n$ and $m_n$ respectively represent the maximum and minimum value of financial subsidies or tax incentives in the three major industries. Furthermore, in order to calculate the final level of industrial policy, it is necessary to calculate the standard Euclidean distance ($Y_m$ and $Y_M$) between Y of any point on the Cartesian space of the two types of industrial policy and its lowest and highest points on the following Eq (21):

$$\begin{cases} Y_m = \sqrt{\dfrac{Fs^2_n + Tp^2_n}{Q^2_{Fs} + Q^2_{Tp}}} \\ Y_M = 1 - \sqrt{\dfrac{(Q_{Fs} - Fs_n)^2 + (Q_{Tp} - Tp_n)^2}{Q^2_{Fs} + Q^2_{Tp}}} \end{cases} \tag{21}$$

Finally, the average value of the two is used to determine the final level of industrial policy, as shown in the following Formula (22):

$$Ip = (Y_m + Y_M)/2 \tag{22}$$

Where, *Ip* represents the final level of industrial policy, the larger the value, the higher the level of industrial policy.

**4.3.2. Measurement of high-quality development of foreign trade.** This paper first excavates the annual reports of China's A-share listed enterprises, and selects keywords such as "import and export" to determine the enterprises involved in foreign trade development. Then, by referring to Song Min et al., (2021), the OP method is used to estimate the high quality development level of China's foreign trade (*Fthd*) [19].

$$lnFthd_{i,t} = d_0 + d_1 lnK_{i,t} + d_2 lnL_{i,t} + d_3 lnM_{i,t} + d_4 Age_{i,t} + d_5 Soe_{i,t} + \alpha_i + \beta_t + \lambda_u + \varepsilon_{i,t} \quad (23)$$

In the above Eq (23), $LnFthd_{i,t}$ represents the high-quality development level of foreign trade of enterprise $i$ in $t$ years, and $LnK_{i,t}$ represents the labor input of enterprise $i$ in $t$ years, expressed by the number of employees of enterprise; $LnL_{i,t}$ represents the capital input of $i$ enterprise in $t$ years, expressed by the outward direct investment of the enterprise; $LnM_{i,t}$ represents the intermediate input of enterprise $i$ in $t$ years, expressed by the total export amount minus the tariff, labor compensation and operating surplus; *Age* is the age of enterprise, expressed by $t$ minus the year of establishment; *Soe* is the dummy variable of state-owned enterprise, if it is state-owned enterprise, it is 1; otherwise, it is 0; $\alpha_i$, $\beta_t$, and $\lambda_u$ represent firm, time, and industry fixed effects, respectively, while $\varepsilon_{i,t}$ represent random disturbance terms.

Concurrently, text mining techniques are utilized in this study to examine the occurrence of words related to the enhancement of foreign trade's quality development, aiming to minimize plagiarism detection rates. Drawing upon the research findings of Wang Min (2022), a comprehensive approach is taken in selecting 33 three-level indicators that encompass various facets [35]. From the multidimensional connotation of the development of high quality trade comprehensive strength, innovation and development level, open circulation ability, open trade cooperation and trade security system five dimensions, selected 33 level 3 indicators, for the construction of China's foreign trade high quality development level of evaluation index system (specific index system as shown in S1 Appendix). Additionally, The enhancement of foreign trade with a focus on superior quality (denoted as *Fthd1*) is formulated through the application of methods (19) to (22).

In addition, since the index measurement of industrial policy and high-quality development of foreign trade has been divided into industries and industries, that is, the fixed effect of enterprises is controlled, then only time fixed effect $\beta_t$ needs to be considered in the empirical process, as shown in Eq (24) below, which also improves the traditional method of controlling individual fixed effect.

$$Fthd_{i,t} = a_0 + a_1 Policy_i \times Treat_t + a_2 X_{i,t} + \beta_t + \varepsilon_{i,t} \quad (24)$$

**4.3.3. Measure of other variables.** The high-quality development of foreign trade is significantly influenced by industrial policy, which affects both the size of enterprises and technological advancements, as indicated in the theoretical analysis of this study. To measure enterprise size, this research adopts a similar approach as Ali and Johl (2023) by utilizing the logarithm of total assets [36]. A higher value indicates a larger enterprise scale (Ta). Additionally, a robustness test is conducted using the cash flow ratio (Cfr), which is calculated as net cash flows from operating activities divided by total assets. Furthermore, following Ye and Ying's (2023) methodology, technological level (Ri) is assessed based on R&D investment made by enterprises [37]. A higher value signifies greater emphasis on technological innovation. The robustness test also includes examining the number of patent applications (Npa), representing the annual count of invention patents and utility model patents applied for.

**Table 1. Descriptive statistical results of each variable.**

| Variable | Symbol | N | mean | sd | min | max |
|---|---|---|---|---|---|---|
| High-quality development of foreign trade | Fthd | 15,974 | 15.7963 | 3.8875 | 0.0001 | 20.7082 |
| High-quality development of foreign trade -1 | Fthd1 | 15,974 | 0.0011 | 0.0068 | 0.0001 | 0.2369 |
| Policy subsidy | Fs | 15,974 | 1.6540 | 5.3768 | 0.0001 | 109.7000 |
| Tax incentives | Tp | 15,974 | 0.1302 | 0.1912 | 0.0001 | 0.8376 |
| Industrial policy | Ip | 15,974 | 0.1593 | 0.1930 | 0.0001 | 0.8847 |
| Total assets | Ta | 15,974 | 22.7832 | 1.5905 | 19.0456 | 31.3101 |
| Cash flow ratio | Cfr | 15,974 | 0.0514 | 0.0757 | -0.7443 | 0.7713 |
| R&d investment | Ri | 15,974 | 0.1354 | 0.4526 | 0.0001 | 18.1959 |
| Number of patents | Npa | 15,974 | 0.0688 | 0.4152 | 0.0001 | 14.9190 |
| Current ratio | Fr | 15,974 | 1.9489 | 2.8769 | 0.0001 | 204.7420 |
| Proportion of fixed assets | Pfa | 15,974 | 0.2270 | 0.1805 | 0.0002 | 0.9709 |
| Listed years | Ly | 15,974 | 2.6427 | 0.5511 | 0.0001 | 3.4965 |
| Number of directors | Nd | 15,974 | 2.1771 | 0.2086 | 1.0986 | 3.0445 |
| Corporate leverage ratio | Clr | 15,974 | 0.4861 | 0.2026 | 0.0071 | 2.0327 |
| Return on equity | Roe | 15,974 | 0.0769 | 0.2174 | -22.1201 | 1.5363 |

Moreover, control variables are selected based on Hipolito et al., (2023) article as well as Ma Tao et al., (2023) work [38, 39]. These variables include current ratio (*Fr*: current assets divided by current liabilities), fixed assets ratio (*Pfa*: net fixed assets divided by total assets), listing time in years minus incorporation year(*Ly*), number of directors represented in logarithmic form (*Nd*: logarithm of board members), corporate leverage ratio(*Clr*: liabilities divided by total assets), return on equity(*Roe*: Net profit divided by average balance of shareholders' equity). Descriptive statistical results for these variables are presented in Table 1. It should be noted that although some values may appear as zero in Table 1.

## 4.4. Correlation analysis of main variables

The present study investigates the relationship between key variables, and the specific findings are presented in Table 2. First of all, the maximum correlation coefficient of the main variable is 0.400, that is, the possibility of more serious multicollinearity is small, slight multicollinearity can be taken without action; Secondly, most of the correlation coefficients are significant at the 5% level, that is, the selected variables are effective; Finally, the correlation coefficients of *Ta* and *Ri*, *Ta* and *Npa*, and *Cfr* and *Npa* are respectively significant 0.040, -0.055 and 0.019, that

**Table 2. Correlation analysis results of main variables.**

|  | Fthd | Fs | Tp | Ip | Ta | Cfr | Ri | Npa |
|---|---|---|---|---|---|---|---|---|
| Fthd | 1 | | | | | | | |
| Fs | 0.027*** | 1 | | | | | | |
| Tp | 0.056*** | 0.018** | 1 | | | | | |
| Ip | 0.021*** | 0.068*** | 0.074*** | 1 | | | | |
| Ta | 0.091*** | 0.027*** | 0.017** | 0.080*** | 1 | | | |
| Cfr | -0.052*** | 0.034*** | 0.400*** | -0.038*** | -0.027*** | 1 | | |
| Ri | 0.107*** | 0.00800 | 0.036*** | -0.074*** | 0.040*** | 0.002 | 1 | |
| Npa | -0.067*** | 0.00200 | -0.028*** | -0.066*** | -0.055*** | 0.019** | 0.012 | 1 |

Note: *, * *, * * * are significant at 10%, 5%, and 1% levels, respectively (same below without special instructions).

is, the variables of the mechanism of action are mutually affected, and such a single or parallel mechanism cannot be simply considered, but it is necessary to combine them.

## 5. Empirical result

### 5.1. Baseline regression result

The specific results of Eq (16) and Eq (24) of the differential differential model are shown in Table 3. Column (1) of Table 3 considers the fixed effect of enterprises, and column (2) adds control variables. Both results prove that industrial policy can significantly promote the *Fthd*. After the current ratio, the proportion of fixed assets, the year of listing, the number of directors, the leverage ratio and the return on equity, the promoting effect is significantly reduced, but the goodness of fit $R^2$ is significantly increased, indicating that the relevant control variables are effective. Besides, the current ratio and the proportion of fixed assets inhibit the *Fthd*, the other variables have significant promoting effects. Columns (3) and (4) take into account the time fixed effect, and their coefficients are larger than those in columns (1) and (2), but their significance remains unchanged, indicating that in the "new round" of China-Us trade friction in 2018 and its aftermath, a 1% increase in industrial policy will significantly promote the *Fthd* by 1.8240%, and with the development of time, the correlation between the superior quality of China's foreign trade and fixed assets will transform from being restrictive to becoming supportive. In column (5) and column (6), both time and firm fixed effect were considered for baseline regression, and the significance of baseline regression was reduced, and the goodness of fit $R^2$ was also reduced, compared with controlling only time fixed effect. Overall, the goodness of Fit $R^2$ for column (4) is relatively best, which means that it is reasonable to review the model in Formula (24) as the baseline for this paper.

### 5.2. Parallel trend test

In order to address potential biases in assessing the impact of industrial policy during the 2018 "new round" of trade frictions between China and the US, this study employs a parallel trend

Table 3. Results of the baseline regression analysis.

| Variable | (1) | (2) | (3) | (4) | (5) | (6) |
|---|---|---|---|---|---|---|
| Ip*Treat | 0.7739*** | 0.2709*** | 2.2343*** | 1.8240*** | 2.2479** | 0.2280** |
| | (8.51) | (2.76) | (8.36) | (6.96) | (2.22) | (2.06) |
| Fr | | -0.0155*** | | -0.0413*** | | -0.0169*** |
| | | (-2.79) | | (-3.47) | | (-3.02) |
| Pfa | | -0.6692*** | | 3.8937*** | | -0.5903*** |
| | | (-4.00) | | (22.56) | | (-3.48) |
| Ly | | 0.4331*** | | 0.1360** | | 0.2246*** |
| | | (11.89) | | (1.97) | | (3.10) |
| Nd | | 0.0542 | | -2.0177*** | | 0.0655 |
| | | (0.49) | | (-13.63) | | (0.59) |
| Clr | | 1.3238*** | | 0.5366*** | | 1.3758*** |
| | | (10.45) | | (3.13) | | (10.74) |
| Roe | | 0.7072*** | | 0.8594*** | | 0.7123*** |
| | | (12.00) | | (6.14) | | (12.06) |
| Time-fixed effect | No | No | Yes | Yes | Yes | Yes |
| Firm fixed effect | No | No | No | No | Yes | Yes |
| Sample size | 15,974 | 15,974 | 15,974 | 15,974 | 15,974 | 15,974 |
| $R^2$ | 0.0049 | 0.0347 | 0.0044 | 0.0474 | 0.0161 | 0.0367 |
| F | 72.45 | 76.15 | 69.85 | 113.46 | 17.36 | 28.25 |

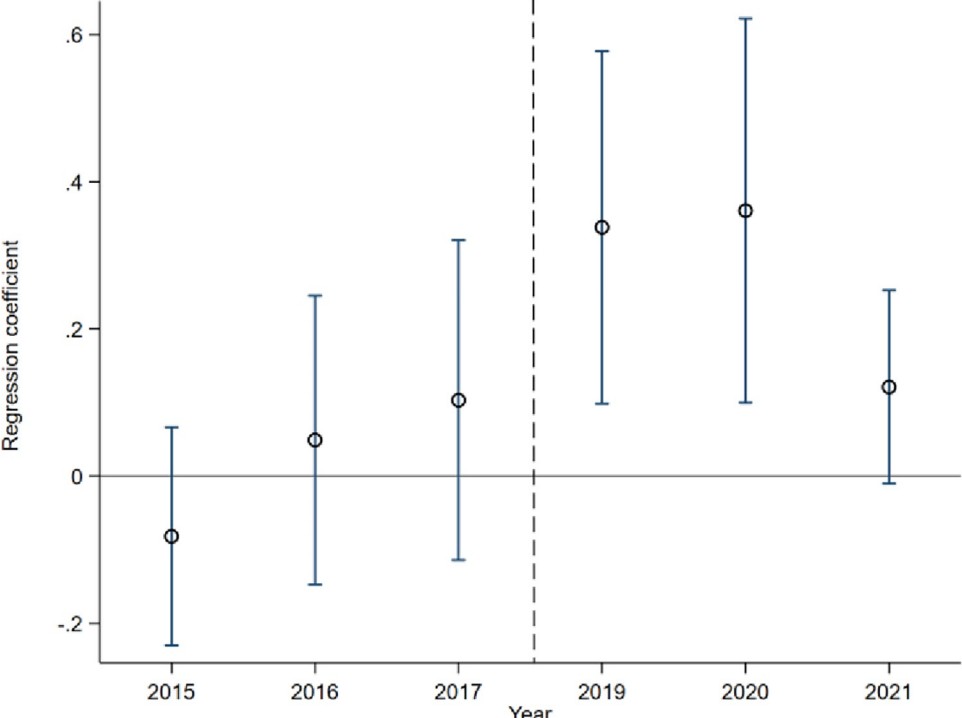

**Fig 2. Parallel trend test.**

analysis. Furthermore, Eq (16) is adjusted by modifying the event time for Treatt while keeping other variables constant. The findings from the parallel trend analysis are displayed in Fig 2. In the three years before 2018, the impact factors of industrial policy were not significant, and the inhibitory effect in 2015 was transformed into the promoting effect in 2016 and after, showing a significant promoting effect from 2019 to 2021, and the impact factors in 2019 were significantly larger than those in 2017. At the same time, due to the impact of the global novel coronavirus epidemic in 2021, although the promotion effect has decreased, it is still significantly larger than the value in 2017.

## 5.3. Robustness test

To ensure the reliability of the baseline regression findings in Eq (16), this study conducts robustness examinations employing three distinct methodologies.

For instance, It is apparent that the eastern provinces have a more sophisticated approach to industrial policy compared to the central and western regions. Therefore, we conducted a benchmark regression analysis in the controlled provinces, as presented in column (1) of Table 4. Furthermore, it was observed that provincial industrial policies exhibited significant positive impacts on Fthd while their influence factors were notably diminished, indicating variations among them.

The second group pertains to the examination of how different indicators chosen may affect empirical results. To tackle this concern, we utilized a text mining technique to substitute China's superior development indicators for international commerce, and these outcomes are displayed in column (2) of Table 4. Although the estimated value is reduced, it is still significant at the 5% level, that is, the higher the level of implementing industrial policies for enterprises, the greater the degree of *Fthd*. On the other hand, financial subsidies and tax incentives are

**Table 4. Robustness test.**

| Variable | (1) | (2) | (3) | (4) | (5) | (6) |
|---|---|---|---|---|---|---|
| | *Fthd* | *Fthd1* | *Fthd* | *Fthd* | *Fthd* | *Fthd* |
| *Ip\*Treat* | 1.7922*** | 0.0011** | | | 1.8076*** | 0.9308*** |
| | (7.13) | (2.40) | | | (6.86) | (3.11) |
| *Fs\*Treat* | | | 0.0268*** | | | |
| | | | (3.69) | | | |
| *Tp\*Treat* | | | | 0.0588 | | |
| | | | | (0.23) | | |
| Control variable | Yes | Yes | Yes | Yes | Yes | Yes |
| Time-fixed effect | Yes | Yes | Yes | Yes | Yes | Yes |
| Provincial fixed effect | Yes | No | No | No | No | No |
| Sample size | 15,974 | 15,974 | 15,974 | 15,974 | 12,551 | 7,644 |
| $R^2$ | 0.1326 | 0.0013 | 0.0453 | 0.0445 | 0.0520 | 0.0379 |
| F | 117.16 | 2.92 | 108.26 | 106.23 | 92.28 | 42.87 |

chosen to represent industrial policies respectively. The findings are displayed in the third and fourth columns of Table 4. A 1% increase in financial subsidies will significantly promote the *Fthd* by 0.0268%, while tax incentives will increase by 0.0588% in value, but it is not significant at the level of 10%. In other words, the effect of foreign trade subsidies implemented by the government is better than that of tax incentives, which indirectly proves that the weight of financial subsidies 0.6889 is greater than that of tax incentives 0.3111 is reasonable when calculating the weight of the two.

The aftermath of the global financial crisis in 2008 had a significant impact on China's economy. Nevertheless, this paper does not take into account the years between 2009 and 2011, which include three years following the crisis. The results are shown in column (5) of Table 4, which still has a significant promoting effect. On the other hand, given that the financial subsidies and tax incentives provided in the industrial policies of provincial capital cities exceed those offered in other cities, this study excludes enterprises situated in provincial capital cities. The corresponding outcomes are presented in column (6) of Table 4. It is important to highlight that industrial policies continue to play a significant role in promoting the *Fthd*.

### 5.4. Assessment of endogeneity

Enterprises in areas with a higher level of high-quality foreign trade development (such as Beijing, Shanghai, Shenzhen, etc.) tend to have strong economic strength and receive higher financial subsidies and tax incentives from the government, that is, there is a reverse causal relationship between industrial policy and high-quality foreign trade development. At the same time, the enterprises with a higher level of foreign trade development in the previous year will maintain a good development trend to a greater extent, and may still achieve a higher level of development in the next year, that is, the high-quality development of foreign trade in the current period will be subtly affected by the high-quality development of foreign trade in the previous period.

In order to avoid the impact of the lag period of industrial policy and high-quality development of foreign trade on the high-quality development of foreign trade in the current period, we use the instrumental variable method to verify whether there is an endogenous problem, and take the lag period of industrial policy as the instrumental variable to conduct 2SLS regression to solve the endogenous problem caused by reverse causality. The specific results are shown in columns (1) and (2) of Table 5. At the same time, columns (3) and (4) take the lag

**Table 5. Endogeneity test.**

| Variable | (1) | (2) | (3) | (4) | (5) | (6) |
|---|---|---|---|---|---|---|
| | The first stage | The second stage | The first stage | The second stage | The first stage | The second stage |
| Ip*Treat | | 7.0536** | | 557.8381*** | | 25.8083*** |
| | | (6.27) | | (6.23) | | (18.86) |
| L_Ip*Treat | 0.2607*** | | | | 0.2584*** | |
| | (29.67) | | | | (29.39) | |
| L_Fthd | | | 0.0016*** | | 0.0012*** | |
| | | | (6.22) | | (4.81) | |
| Control variable | Yes | Yes | Yes | Yes | Yes | Yes |
| Time-fixed effect | Yes | Yes | Yes | Yes | Yes | Yes |
| Sample size | 14833 | 14,833 | 14,833 | 14,833 | 14,833 | 14,833 |
| Wald F | 880.57 | | 38.73 | | 452.49 | |
| Prob > F | | 0.0000 | | 0.0000 | | 0.0000 |

period of high-quality development of foreign trade as an instrumental variable to solve the second problem. Columns (5) and (6) consider that a period of lag in industrial policy and a period of lag in high-quality development of foreign trade may have simultaneous effects. Wald F-values for the weak instrumental variable tests in columns (1), (3), and (5) are all greater than 10. It means that the instrumental variables (the lag phase of industrial policy and the lag phase of high-quality development of foreign trade) are related to the explanatory variable (the industrial policy of the current period), and are not weak instrumental variables. Meanwhile, the results of Prob > F in the remaining three columns are all less than 10%, indicating rejection of the null hypothesis, and the lag phase of high-quality development of industrial policy and foreign trade is an exogenous variable. There is no such endogeneity problem.

## 5.5. Heterogeneity test

The industrial policies of enterprises in different property rights, strategic cities and geographical regions may have different impacts on the *Fthd*, so the heterogeneity test is divided into the following three categories:

**5.5.1. Firm heterogeneity test.** State-owned enterprises play a pivotal role in the national economy, often receiving government support for their foreign trade activities. They serve as the primary impetus behind the advancement of high-quality foreign trade.Although private enterprises have stronger flexibility and innovation ability, and can adapt to the changes of the international market faster, they have difficulty in financing and weak ability to cope with the impact of international risks, and move slowly in the process of high-quality foreign trade development. Based on the classification of enterprises according to their property rights, the sample companies are divided into state-owned and private entities. From the data presented in columns (1) and (2) of Table 6, it becomes apparent that industrial policies implemented by state-owned enterprises have a more pronounced impact on promoting Fthd compared to those adopted by private enterprises. This observation implies that state-owned enterprises demonstrate greater resilience in effectively managing risk shocks amidst "new round" of Sino-US trade friction in 2018.

**5.5.2. Urban heterogeneity test.** Frequent trade exchanges between countries or regions need to rely on a series of factors such as highly open trade systems and policies, long-term strategic cooperative relations, convenient transportation conditions, and differentiated product and service structures of both sides. On the basis of these key elements, the "Belt and Road" Initiative provides more foreign trade opportunities for countries or regions along the route

**Table 6. Heterogeneity test.**

| Variable | (1) | (2) | (3) | (4) | (5) | (6) | (7) | (8) |
|---|---|---|---|---|---|---|---|---|
| | State-owned enterprise | Private enterprise | the Belt and Road. | Non-belt and Road | Eastern region | Central region | Western region | Western region |
| $Ip*Treat$ | 2.2802*** | 1.2308*** | 3.7479*** | 1.1135*** | 2.0586*** | 0.9394* | 1.3565* | 3.4358*** |
| | (6.60) | (3.08) | (6.02) | (4.11) | (6.12) | (1.94) | (1.91) | (2.83) |
| Control variable | Yes | Yes | Yes | Yes | Yes | Yes | Yes | Yes |
| Time-fixed effect | Yes | Yes | Yes | Yes | Yes | Yes | Yes | Yes |
| Sample size | 9,226 | 6,748 | 4,522 | 11,452 | 10,248 | 2,548 | 2,408 | 770 |
| $R^2$ | 0.0563 | 0.0479 | 0.1254 | 0.0289 | 0.0881 | 0.0475 | 0.0864 | 0.0864 |
| F | 78.51 | 48.34 | 96.18 | 48.67 | 141.21 | 17.99 | 32.24 | 10.12 |

through mutualism and industrial transfer, so as to achieve global trade prosperity and hedge the adverse effects of trade protectionism to a certain extent. Due to the obvious differences in the development level of various regions in China, the Belt and Road Initiative has clearly defined the opening-up trend of each region in the process of promoting the initiative, and has focused on distinguishing 18 provinces involving the northeast, northwest and southwest regions according to their economic functions and roles. The cities where enterprises are registered can be categorized into two groups in this study: those located within the Belt and Road region, and those situated outside of it. This categorization is conducted as part of an analysis on their association with the "Belt and Road" Initiative.The specific outcomes can be found in the third and fourth columns of Table 6. Despite the lower number of enterprises in cities along the Belt and Road compared to other cities, their industrial policies exert a disproportionately influential role in promoting *Fthd*, suggesting that trade cooperation can effectively amplify this promotional impact.

**5.5.3. Regional heterogeneity test.** Based on the trajectory of economic growth, China can be classified into four primary geographical divisions: the eastern, central, western, and northeastern regions.The eastern region has a large number of port cities and special economic zones, which occupy a dominant position in foreign trade [40], while the central region makes a great contribution to the manufacturing industry such as steel and automobiles and the export of agricultural products [41]. The western region has advantages in the export of natural resources such as oil, natural gas and coal as well as agricultural products [42], while the northeastern region contributes significantly to the foreign trade of heavy industrial products and resource products [43]. The impact of different regional industrial policies on the *Fthd* is shown in columns (5) to (8) of Table 6. The promotion effect of the eastern region is much greater than that of the central and western regions, and the number of enterprises in the northeast region is only 55, but its estimated value is the largest, indicating that after the "new round" of Sino-US trade friction in 2018, the economically developed region is still the key region for the *Fthd*.

## 6. Mechanism analysis

Theoretical analysis shows that industrial policy affects the *Fthd* by acting on enterprise scale and technology. Therefore, this paper combines Formula (17) and Formula (18) to conduct a stepwise regression test.

### 6.1. Mechanism test of scale effect

Column (1) in Table 7 shows the impact of industrial policies on total assets, and its estimated value is significantly negative, which means that under the "new round" of Sino-US trade

**Table 7. Test of mechanism of action.**

| Variable | (1) | (2) | (3) | (4) | (5) | (6) | (7) | (8) |
|---|---|---|---|---|---|---|---|---|
| | *Ta* | *Fthd* | *Cfr* | *Fthd* | *Ri* | *Fthd* | *Npa* | *Fthd* |
| *Ip*Treat* | -0.2406*** | 1.7703*** | 0.0104** | 1.7849*** | -0.0959*** | 1.7785*** | -0.0690** | 1.8338*** |
| | (-2.90) | (6.77) | (2.15) | (6.82) | (-3.08) | (6.79) | (-2.41) | (6.99) |
| *Ta* | | -0.2234*** | | | | | | |
| | | (-8.97) | | | | | | |
| *Cfr* | | | | 3.7462*** | | | | |
| | | | | (8.79) | | | | |
| *Ri* | | | | | | -0.4747*** | | |
| | | | | | | (-7.14) | | |
| *Npa* | | | | | | | | 0.1412* |
| | | | | | | | | (1.95) |
| Control variable | Yes | Yes | Yes | Yes | Yes | Yes | Yes | Yes |
| Time-fixed effect | Yes | Yes | Yes | Yes | Yes | Yes | Yes | Yes |
| Sample size | 15,974 | 15,974 | 15,974 | 15,974 | 15,974 | 15,974 | 15,974 | 15,974 |
| $R^2$ | 0.3700 | 0.0522 | 0.1256 | 0.0520 | 0.0082 | 0.0505 | 0.0015 | 0.0476 |
| F | 1338.37 | 109.82 | 327.33 | 109.41 | 18.85 | 105.95 | 3.51 | 99.77 |
| Sobel Z | | 2.755*** | | 2.088** | | 2.83*** | | -1.52 |

friction in 2018, more financial subsidies and tax incentives for enterprises by the government are not conducive to the increase of total assets of enterprises, and inhibit the economic development potential of enterprises in international shocks. It may be due to the "new round" Sino-US trade impact, the higher threshold of financial subsidies and tax incentives, stricter supervision and stronger market competition constraints, etc. inhibit the enthusiasm of enterprises to increase assets [44]; It can be seen that the influence of industrial policy and total assets on the *Fthd* in column (2) has passed the Sobel test, and the estimated value of c1 in Eq (17) is lower than 1.8240 in column (4) of Table 3, that is, there is a single mechanism of action. The promotion effect of industrial policy on the *Fthd* will be partly offset by the inhibition effect of total assets on the *Fthd*. It may be that the increase in total assets of enterprises under the "new round" Sino-US trade friction leads to overcapacity, the urgent pursuit of short-term profits and the reduction of product costs, and the lack of management ability inhibit the *Fthd* [45].

The results presented in columns (3) and (4) of Table 7 demonstrate that the implementation of industrial policies has a positive impact on enterprises' cash flow ratio. However, it is noteworthy that the promoting effect of industrial policies on the *Fthd* is overshadowed by the enhancing influence of enterprises' cash flow ratio on the *Fthd*, which may be attributed to an increase in financial subsidies and post-tax incentives since 2018. In addition, the increase of cash flow will promote enterprises to carry out foreign trade investment, technological innovation and market competition and other activities, but when enterprises' foreign trade activities increase, the government's financial subsidies will decrease and the tax will increase, which will have a crowding out effect on the *Fthd* [46].

## 6.2. Mechanism test of technology effect

Columns (5) and (6) of Table 7 indicate that industrial policy significantly inhibits enterprises' R&D input, and the promotion effect of industrial policy on *Fthd* will be offset by the inhibition effect of enterprise R&D on *Fthd*. It may be due to the increase of financial subsidies and tax incentives given by the government after 2018, which makes relevant enterprises more

inclined to pursue the support of government subsidies. Reduce R&D expenditure to maintain survival, and even if the increase in R&D investment promotes the increase in product cost and profit, the direct impact of Sino-US trade friction is the decrease in sales of related products, and these problems inhibit the *Fthd* [47, 48].

The influence of industrial policy on the Fthd is affected by the number of patent applications made by enterprises in column (7) and List (8) of Table 7. Although the significance level of the Sobel test does not meet the threshold, it can be observed that an increase in enterprise patent applications enhances the promotional effect of industrial policy on the *Fthd*. This also means that after the "new round" of Sino-US trade friction in 2018, Chinese enterprises will break through the technological blockade of the United States to help promote the *Fthd* [49].

### 6.3. Further discussion

In Table 2, the correlation between total assets and R&D input is significant, and in Table 7, total assets and R&D input have a single significant mechanism of action. In the analysis of the appellate theoretical framework, the scale effect and technology effect of enterprises jointly influence the promotion effect of industrial policy on the *Fthd*. Therefore, we combined Formula (16), Formula (17) and Formula (18) to test the interaction mechanism of enterprise scale effect and technology effect.

First of all, Table 7 column (5), (1) and Table 8 (1) of the results of the total assets effective effect on the influence of the industry on r & d investment, namely industrial policy on r & d investment (0.0959***) by the industrial policy on total assets (0.2406***) and reduced to 0.0940, may be due to the increase of the enterprise assets will make enterprises have enough capital to increase r & d investment, improve the international competitiveness of products under the "new round of" trade friction [50].

Secondly, the results of columns (1) and (5) of Table 7 and (2) of Table 8 show that the total assets of industrial policy on R&D input (-0.2406***) will be reduced to -0.2353 by the inhibitory effect of industrial policy on R&D input (-0.0959***). This may be due to the increase in R&D investment of enterprises, which reduces product costs, increases sales, and increases total assets of enterprises, alleviating the inhibitory effect of industrial policy on R&D investment [51].

Finally, by comparing the estimated value of c1 1.7268 and 1.8240, 1.7703 and 1.7785 in column (3) of Table 8, column (4) of Table 7, column (2) of Table 7 and column (6) of Table 7, we can see that the scale effect and technology effect of enterprises interact with the promotion effect of industrial policy on the *Fthd*. A more accurate result of Fig 1 is shown in Fig 3 below.

### 7. Conclusions and enlightenments

Following the "new round" of Sino-US trade friction in 2018, it is imperative to evaluate the efficacy of China's industrial policy in promoting high-quality development in foreign trade (Fthd) and its underlying mechanisms. This study establishes a theoretical framework based on Kogangras' production function to analyze this issue. It conducts both qualitative and quantitative analyses using financial data from 1141 Chinese A-shares listed companies between 2009 and 2022. The empirical results indicate that for every 1% increase in the intensity of government-implemented industrial policy, there is a corresponding enhancement of 1.8240% in the extent of China's foreign trade characterized by superior development quality. Importantly, additional tests considering factors such as provincial fixed effects, replacement variables, and extreme values consistently confirm this significant promotional effect. This effect is particularly pronounced for enterprises located in regions with higher policy intensity, more frequent trade exchanges, and greater economic development. Furthermore, three

**Table 8. Interaction mechanism test.**

| Variable | (1) | (2) | (3) |
|---|---|---|---|
| | *Ri* | *Ta* | *Fthd* |
| *Ip\*Treat* | -0.0940*** | -0.2353*** | 1.7268*** |
| | (-3.04) | (-2.83) | (6.61) |
| *Ta* | 0.0078** | | -0.2197*** |
| | (2.64) | | (-8.83) |
| *Ri* | | 0.0558** | -0.4624*** |
| | | (2.64) | (-6.97) |
| Control variable | Yes | Yes | Yes |
| Time-fixed effect | Yes | Yes | Yes |
| Sample size | 15,974 | 15,974 | 15,974 |
| $R^2$ | 0.0086 | 0.3703 | 0.0551 |
| F | 17.37 | 1172.39 | 103.30 |
| Sobel Z | -1.95* | -2.01** | |

scenarios are identified: Firstly, industrial policies may either hinder total asset growth or facilitate an increase in cash flow ratio for enterprises resulting in a diminished promotional effect on Fthd; Secondly restrictions on R&D investment imposed by industrial policy can impede Fthd promotion; However limitations on patent applications can promote Fthd without significant evidence supporting this mechanism; lastly the interplay between total enterprise assets and R&D investment attenuates the promotional impact of industrial policy on Fthd.

Based on the findings, the paper derives three valuable lessons for future research and policy deliberations.

## (1) Accelerating the transformation of enterprises into high-end entities

In the current market environment, the high-end transformation of enterprises includes three aspects: digital, intelligent and green:

Within the framework of corporate digital transformation, businesses are accelerating the integration of digital technology across various facets of their activities, including research and development, procurement, production, warehouse management, marketing, and other business processes. This strategic endeavor aims to optimize organizational structure, refine management practices, and enhance governance models through leveraging digital technology. Moreover, there is a heightened emphasis on investing in research and development activities to fully harness the internal capacity-building benefits facilitated by digital transformation. Enterprises should give full play to the advantages of digital platforms such as the Internet, screen complex market information, tap potential investment opportunities, and rely on the characteristics of digital technology openness and sharing and breaking time and space

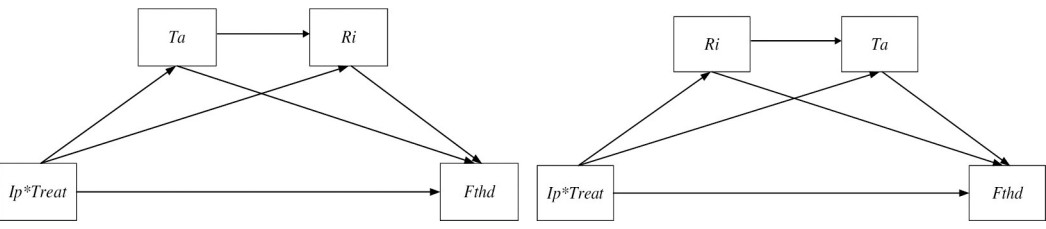

**Fig 3. Interaction mechanism diagram.**

restrictions, efficiently and accurately handle massive digital resources in the market, accurately identify the increasingly diversified needs of customers, open up new customer resource markets, and thus significantly reduce external transaction costs. Improve production efficiency.

In terms of intelligent transformation of enterprises, given that the integration of various organizational management, production activities and intelligent technology will generate collaborative costs, moderate intelligent investment will help enterprises develop high-quality, but excessive intelligent investment will form a crowding out effect with other production factors, greatly increasing the overall collaborative cost and hindering the improvement of production efficiency. Therefore, enterprise managers should not only fully realize the importance of intelligence, but also formulate the quality layer design to accelerate the intelligent transformation of the whole process of the enterprise from the strategic level, and expand the investment in the integration of intelligent hardware and software resources.

Enterprises must take proactive responsibility in efficiently managing and reducing environmental impacts within the framework of corporate environmental transformation.This can be accomplished by establishing a robust environmental management information system, promptly disclosing comprehensive details regarding their environmental governance efforts, and garnering the trust and support of market stakeholders through tangible actions towards green transformation. By doing so, companies can bolster their reputation and long-term market value. It is crucial to acknowledge that green transformation encompasses not only environmental performance but also economic performance and sustainability innovation. Therefore, regular self-assessment should be conducted to gauge the progress made in achieving sustainable development goals. Identifying key obstacles impeding green transformation is essential for actively integrating a mindset of green development into all aspects of production and operations. Moreover, leveraging environmentally-friendly technologies will facilitate a seamless transition towards a greener future by promoting resource efficiency. Ultimately, these measures will contribute significantly to successfully attaining the "double carbon" goal.

## (2) Reinforce the exploitation of the extensive market to facilitate the advancement of industrial policies

China has the largest consumer market in the world, which gives it a unique advantage in dealing with external pressures. The Chinese government continues to promote the continuous expansion of the domestic market through measures such as expanding domestic demand and promoting consumption upgrading. It is also necessary to encourage more enterprises to participate in the domestic market and reduce their dependence on external markets by stimulating consumption and improving the development of the service sector.

First of all, analyze the market demand and trend, through market research and data analysis, understand the demand and trend of the domestic market, and determine the priority development of industrial areas. Taking into account the characteristics of the hyperscale market, focus on industries with high growth potential, high added value, high technology content and conducive to domestic consumption. By stimulating the vitality of the domestic market, strengthening the construction of the domestic market and reducing the dependence on the external market.

Secondly, strengthen the encouragement of innovation and R&D, increase support for scientific and technological advancements in key industries, and encourage businesses to enhance their technical capabilities and ability to innovate independently.The government should set up science and technology innovation funds, support scientific research institutions and enterprises to carry out innovation projects, and provide patent protection and intellectual

property support. At the same time, we will strengthen the training of government departments and enterprises to improve their understanding of international trade policies and market changes, and cultivate professionals with an international perspective and policy analysis ability.

Finally, establish standards and quality systems that are in line with the international market, and enhance the international competitiveness of priority industries by strengthening quality supervision, standard setting and brand building. By supporting enterprises to participate in international exhibitions and trade negotiations, the government cultivates internationally renowned brands, promotes the international market expansion of priority industries, and improves the competitiveness of domestic products.

## (3) Enhance the ability to respond to foreign trade policy implementation

In the context of the "new round" of Sino-US trade friction, we must take measures from the perspective of industrial policy to deal with it. Establishing an industrial policy evaluation system to cope with trade protection, improving the industrial policy governance mechanism, and improving policy enforcement will help strengthen the response to trade protectionism, safeguard national interests, and achieve sustainable development.

First, improve the industrial policy evaluation system in response to trade protection. China should further improve its industrial policy evaluation system to better cope with the challenges brought by trade protectionism. These include: Impact assessment: When formulating new industrial policies or making major adjustments, the government will conduct a comprehensive impact assessment to analyze the international trade reaction that the policies may cause, so as to avoid unnecessary trade disputes; Risk assessment: for industries that may be affected by trade protectionist measures, anticipate their possible losses and impacts, and provide coping strategies for enterprises; Policy coordination assessment: there may be conflicts and overlaps between different industrial policies, and policy coordination assessment can ensure the consistency and coordination of industrial policies.

Second, optimize the foreign trade policy feedback mechanism. Effective feedback on policy implementation is the key to dealing with trade protectionism, and China should take a series of measures to improve policy implementation. Specifically include: supervision and inspection: the government will strengthen the supervision and inspection of the implementation of industrial policies to ensure that the policies take root; Accountability mechanism: The government will establish an accountability mechanism for relevant departments and responsible persons who fail to effectively implement industrial policies to ensure the effective implementation of policies; Information disclosure: The government will disclose the implementation of industrial policies in a timely manner, so that enterprises and the public can understand the effect and impact of the policies.

In the end, enhanced global collaboration is of utmost importance. Given the prevailing surge in trade protectionism, China has proactively engaged in international cooperation and jointly confronted the challenges at hand. Concrete actions include safeguarding the integrity of the multilateral trading system, strengthening partnerships with other nations to foster advancements in free trade, actively promoting and entering into free trade agreements to broaden our network of trading allies and reduce barriers to commerce, as well as establishing a consultation mechanism that facilitates dialogue with relevant countries for addressing potential issues leading to trade disputes and resolving differences amicably.

However, there are still three shortcomings in this study.first, Limitations to the study data exist, More businesses in 2009 and 2022 and we deleted, at the same time, The enterprise data screened in this paper were largely missing before 2008 and after 2023, After expanding the

number of enterprises or the study period, Whether the results remain consistent; next, The choice of the model is too monotonous, The dual difference model is more suitable for the discussion of this paper, But whether the conclusions of the remaining regression models are similarly similar are unclear, For example, the space Dubin model to explore the impact of industrial policies of surrounding provinces on the high-quality development of foreign trade of enterprises in the provinces; last, The validation of the model assumptions cannot be comprehensive, For example, adopting the Levinsohn-Petrin method or the ordinary least squares method to measure the high-quality development of foreign trade and adopting the entropy method to measure the industrial policy level, Making the conclusions more robust. Therefore, the issue of appeal can be further refined in future studies.

## Supporting information

**S1 Appendix.**
(DOCX)

## Author Contributions

**Conceptualization:** He Leihua, Sun Fan.

**Formal analysis:** Sun Fan.

**Investigation:** He Leihua.

**Methodology:** He Leihua, Sun Fan.

**Writing – original draft:** He Leihua, Sun Fan.

**Writing – review & editing:** He Leihua, Sun Fan.

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
