## [Decision Letter · Decision Letter 0]

29 Jul 2024

PONE-D-24-27007Empirical Evidence on the Impact of the "New Round" of Sino-US Trade Frictions on China's Foreign Trade Industrial Policy and High-quality DevelopmentPLOS ONE

Dear Dr. fan,

Thank you for submitting your manuscript to PLOS ONE. After careful consideration, we feel that it has merit but does not fully meet PLOS ONE’s publication criteria as it currently stands. Therefore, we invite you to submit a revised version of the manuscript that addresses the points raised during the review process.

We look forward to receiving your revised manuscript.

Kind regards,

Guangnian Xiao

Academic Editor

PLOS ONE

Journal Requirements:

3. For studies involving third-party data, we encourage authors to share any data specific to their analyses that they can legally distribute. PLOS recognizes, however, that authors may be using third-party data they do not have the rights to share. When third-party data cannot be publicly shared, authors must provide all information necessary for interested researchers to apply to gain access to the data. (https://journals.plos.org/plosone/s/data-availability#loc-acceptable-data-access-restrictions) 

Additional Editor Comments:

Please revise the manuscript according to comments from two reviewers.

Reviewers' comments:

Reviewer's Responses to Questions

**Comments to the Author**

1. Is the manuscript technically sound, and do the data support the conclusions?

Reviewer #1: Partly

Reviewer #2: Partly

2. Has the statistical analysis been performed appropriately and rigorously? 

Reviewer #1: No

Reviewer #2: I Don't Know

3. Have the authors made all data underlying the findings in their manuscript fully available?

Reviewer #1: No

Reviewer #2: No

4. Is the manuscript presented in an intelligible fashion and written in standard English?

Reviewer #1: No

Reviewer #2: No

5. Review Comments to the Author

Reviewer #1: 1. Whether the small values in Table 1 are written as 0 will not affect the result?Please give an explanation.

2. In Table 2 ,why any concerns regarding multicollinearity can be disregarded for subsequent analyses?

3. In Table 4, there is only a column of the provincial industrial policies, Are the conclusions reliable with one data? Please give a brief explanation.

4. The following studies were recommended to be properly cited: [1] Spatial-Temporal Ship Pollution Distribution Exploitation and Harbor Environmental Impact Analysis via Large-Scale AIS Data. [2] Orientation-aware ship detection via a rotation feature decoupling supported deep learning approach. [3] Technical and economic analysis of battery electric buses with different charging rates. [4] Manufacturers' Emission-reduction Investments in Competing Supply Chains with Prisoner’s Dilemma: The Economic and Environmental Impacts of Retailer(s) Capital Constraint(s)

Reviewer #2: the submission titled "Empirical Evidence on the Impact of the 'New Round' of Sino-US Trade Frictions on China's Foreign Trade Industrial Policy and High-quality Development" is the presence of significant issues in both the methodology and data analysis sections. Specifically:

Methodological Flaws:

The methodology section lacks clarity in defining key variables and their measurements. For instance, the paper mentions using financial subsidies and tax incentives as indicators for industrial policy but does not provide sufficient detail on how these indicators were derived or validated.

The use of panel data from 2009 to 2022 comprising 1141 Chinese A-share listed enterprises is mentioned, but the criteria for selecting these enterprises and the robustness of this dataset are not adequately addressed.

Data Analysis and Interpretation Issues:

The empirical results claim a significant promotion effect of industrial policy on high-quality foreign trade development by 1.8240%. However, the statistical methods used to arrive at this figure are not thoroughly explained, raising concerns about the validity of these findings.

There are inconsistencies in the reported regression results. For instance, Table 3 shows varying results for the impact of industrial policy on foreign trade development, which suggests potential issues with the model specifications or data handling.

Insufficient Addressing of Endogeneity:

While the paper attempts to address endogeneity issues using instrumental variable approaches, the instruments chosen (e.g., lagged values) may not be strong or valid instruments. The justification for their selection and the tests confirming their validity are not adequately discussed.

Lack of Robustness Checks:

The robustness tests presented are limited and do not cover all potential sources of bias. For example, the paper could benefit from additional robustness checks, such as sensitivity analyses or alternative model specifications, to confirm the stability of the results.

Inadequate Discussion of Limitations:

The paper does not sufficiently discuss the limitations of the study. Potential issues such as data limitations, model assumptions, and external validity are not adequately addressed, which could undermine the credibility of the findings.

6. PLOS authors have the option to publish the peer review history of their article (what does this mean?). If published, this will include your full peer review and any attached files.

Reviewer #1: No

Reviewer #2: No

---

## [Author Response · Author response to Decision Letter 0]

6 Aug 2024

Response to Reviewer 1 Comments:

Point 1. Whether the small values in Table 1 are written as 0 will not affect the result?Please give an explanation.

 Author response to Point 1: The minimum value of table 1 appeared "0", mainly due to for the convenience of showing the results, we take rounding in Table 1, if the fifth is less than “5” will appear the phenomenon of "0.0000", and in the actual process, and in the beginning of the screening process of 1% tail processing, namely there is no value for "0" data, so this does not affect the results of the experiment, of course, in order to avoid differences in understanding, we keep the four decimal behind "0" as "0.0001"

Point 2.In Table 2 ,why any concerns regarding multicollinearity can be disregarded for subsequent analyses? 

 Author response to Point 2: In the correlation analysis, if the coefficient of two variables is between 0 and 0.5, it is considered to have less serious multicollinearity, and if more serious multicollinearity is between 0.5-1. However, the maximum value between the coefficient of these variables is 0.400, that is, the possibility of serious multicollinearity is low in this text. At the same time, multicollinearity is common, slight multicollinearity can be taken without action, that is, reasonable error. In order to avoid the previous statement to be too absolute, we changed the sentence above Table 2, "We can ignore the problem of multicollinearity in the following analysis" to "the maximum correlation coefficient of the main variable is 0.400, that is, the possibility of more serious multicollinearity is small, slight multicollinearity can be taken without action".

Point 3. In Table 4, there is only a column of the provincial industrial policies, Are the conclusions reliable with one data? Please give a brief explanation.

 Author response to Point 3: The industrial policy selected in this paper is synthesized by financial subsidies and tax incentives through a series of operations (specific formula 19 and 20), table 4 robust test, column (3) and (4) is the fiscal subsidies and tax incentives (16), to avoid misunderstanding, we added four lines in table 4, the column (3) and (4) column variables in Fs*Treat and Tp*Treat respectively, to avoid the unified use of Ip*Treat and misunderstanding.

Point 4. The following studies were recommended to be properly cited: [1] Spatial-Temporal Ship Pollution Distribution Exploitation and Harbor Environmental Impact Analysis via Large-Scale AIS Data. [2] Orientation-aware ship detection via a rotation feature decoupling supported deep learning approach. [3] Technical and economic analysis of battery electric buses with different charging rates. [4] Manufacturers' Emission-reduction Investments in Competing Supply Chains with Prisoner’s Dilemma: The Economic and Environmental Impacts of Retailer(s) Capital Constraint(s)

 Author response to Point 4: Referring to the appellate literature, we cite it and discuss the inadequacy of this article at the end of the article：However, there are still three shortcomings in this study.first, Limitations to the study data exist, More businesses in 2009 and 2022 and we deleted, at the same time, The enterprise data screened in this paper were largely missing before 2008 and after 2023, After expanding the number of enterprises or the study period, Whether the results remain consistent; next, The choice of the model is too monotonous, The dual difference model is more suitable for the discussion of this paper, But whether the conclusions of the remaining regression models are similarly similar are unclear, For example, the space Dubin model to explore the impact of industrial policies of surrounding provinces on the high-quality development of foreign trade of enterprises in the provinces; last, The validation of the model assumptions cannot be comprehensive, For example, adopting the Levinsohn-Petrin method or the ordinary least squares method to measure the high-quality development of foreign trade and adopting the entropy method to measure the industrial policy level, Making the conclusions more robust. Therefore, the issue of appeal can be further refined in future studies.

Response to Reviewer 2 Comments:

Point 1. The methodology section lacks clarity in defining key variables and their measurements. For instance, the paper mentions using financial subsidies and tax incentives as indicators for industrial policy but does not provide sufficient detail on how these indicators were derived or validated.

 Author response to Point 1: In order to more clearly represent the source and operational process of industrial policy, We have refined the measure part of the industrial policy in the article and marked it red: this article selects the national Taian database, In terms of the logarithm of the production subsidy provided by the government to the enterprises, The tax incentives are indicated by the sum of the taxes returned by the government divided by the taxes paid by the enterprise and the taxes returned received (see Table 1 for specific results), Companies are usually accompanied by fiscal subsidies and tax incentives, Thus referring to the existing literature, The coefficient of variation method can be adopted to determine the weight, The average value of the standard Euclidean distance of any point in Cartesian space and its lowest and highest point of space is calculated by three industries to indicate the industrial policy level of each enterprise.

Point 2. The use of panel data from 2009 to 2022 comprising 1141 Chinese A-share listed enterprises is mentioned, but the criteria for selecting these enterprises and the robustness of this dataset are not adequately addressed.

 Author response to Point 2:About enterprise data screening criteria and robustness in the following description and standard red: In this paper, in 2009 to 2022 China a-share listed companies as the research object (before 2008 and 2023 after data missing, cannot provide reliable experimental results), core data from the Tai'an (CSMR) database, Wind database and China city statistical yearbook, etc., to cross check data and 1% tail processing, and select the research period has been normal operation of the listed companies, finally screening meet the conditions of 1048 China a-share listed companies. 

Point 3. The empirical results claim a significant promotion effect of industrial policy on high-quality foreign trade development by 1.8240%. However, the statistical methods used to arrive at this figure are not thoroughly explained, raising concerns about the validity of these findings.

 Author response to Point 3:This paper concludes that the method for this numerical value is a two-fold difference model, To ensure that the two-fold difference model is valid, one side, We never control the time fixed effect or enterprise fixed effect to the control time and enterprise fixed effect, can find that the goodness of fit of the model R2 is constantly improving, Of the best results in column (4), Namely, a two-fold difference model controlling time-fixed effects is most effective in the existing discussion, That is, the choice (24) is more reasonable than the choice (16); On the other hand, Our practice of referring to the existing literature, Taking a parallel trend test, You can see that in Figure 2, the results in the years after the implementation of the policy are above the horizontal line, That is, the impact of industrial policy on the high-quality development of foreign trade was significantly impacted by the trade friction in 2018, It is effective to adopt a two-fold difference model controlling for the fixed effects of time.

Point 4. There are inconsistencies in the reported regression results. For instance, Table 3 shows varying results for the impact of industrial policy on foreign trade development, which suggests potential issues with the model specifications or data handling.

 Author response to Point 4:First, there is a small error in column (4) of Table 3, where the enterprise fixed effect position should be "No". Second, in table 3 the influence of industrial policy on the development of high quality foreign trade although the numerical deviation (this is because we take different fixed effect in the double difference model and whether to join the control deviation caused by variables), but the symbols are positive and significantly related at 5% level, which means that the industrial policy is significantly promote the development of China's foreign trade high quality (as shown in table 3 in the second line).

Point 5. While the paper attempts to address endogeneity issues using instrumental variable approaches, the instruments chosen (e.g., lagged values) may not be strong or valid instruments. The justification for their selection and the tests confirming their validity are not adequately discussed.

 Author response to Point 5:We added the reasons for choosing industrial policies and the lag period of high-quality foreign trade in the part of endogenous inspection and marked it red: Enterprises in areas with a higher level of high-quality foreign trade development (such as Beijing, Shanghai, Shenzhen, etc.) tend to have strong economic strength and receive higher financial subsidies and tax incentives from the government, that is, there is a reverse causal relationship between industrial policy and high-quality foreign trade development. At the same time, the enterprises with a higher level of foreign trade development in the previous year will maintain a good development trend to a greater extent, and may still achieve a higher level of development in the next year, that is, the high-quality development of foreign trade in the current period will be subtly affected by the high-quality development of foreign trade in the previous period. In order to avoid the impact of the lag period of industrial policy and high-quality development of foreign trade on the high-quality development of foreign trade in the current period, we use the instrumental variable method to verify whether there is an endogenous problem, and take the lag period of industrial policy as the instrumental variable to conduct 2SLS regression to solve the endogenous problem caused by reverse causality. The specific results are shown in columns (1) and (2) of Table 5. At the same time, columns (3) and (4) take the lag period of high-quality development of foreign trade as an instrumental variable to solve the second problem. Columns (5) and (6) consider that a period of lag in industrial policy and a period of lag in high-quality development of foreign trade may have simultaneous effects.

For the effectiveness of the test is mainly a reference to the existing literature, observe the first stage of the Wald F-value and second Prob> F, Wald F-values for the weak instrumental variable tests in columns (1), (3), and (5) are all greater than 10. It means that the instrumental variables (the lag phase of industrial policy and the lag phase of high-quality development of foreign trade) are related to the explanatory variable (the industrial policy of the current period), and are not weak instrumental variables. Meanwhile, the results of Prob > F in the remaining three columns are all less than 10%, indicating rejection of the null hypothesis, and the lag phase of high-quality development of industrial policy and foreign trade is an exogenous variable. There is no such endogeneity problem.

Point 6. The robustness tests presented are limited and do not cover all potential sources of bias. For example, the paper could benefit from additional robustness checks, such as sensitivity analyses or alternative model specifications, to confirm the stability of the results.

 Author response to Point 6:The robustness test mainly improved model (control province fixed effect), replace explanatory variables and explained variables, delete extreme data (provincial capital city enterprise), which includes the sensitivity analysis and alternative model specification, and the results of the experiment can not contain all the robustness test, only as far as possible to reduce the error.

Point 7. The paper does not sufficiently discuss the limitations of the study. Potential issues such as data limitations, model assumptions, and external validity are not adequately addressed, which could undermine the credibility of the findings.

 Author response to Point 7:At the end of the article, we added the description of the research shortcomings and mark the red, as follows: However, there are still three shortcomings in this study.first, Limitations to the study data exist, More businesses in 2009 and 2022 and we deleted, at the same time, The enterprise data screened in this paper were largely missing before 2008 and after 2023, After expanding the number of enterprises or the study period, Whether the results remain consistent; next, The choice of the model is too monotonous, The dual difference model is more suitable for the discussion of this paper, But whether the conclusions of the remaining regression models are similarly similar are unclear, For example, the space Dubin model to explore the impact of industrial policies of surrounding provinces on the high-quality development of foreign trade of enterprises in the provinces; last, The validation of the model assumptions cannot be comprehensive, For example, adopting the Levinsohn-Petrin method or the ordinary least squares method to measure the high-quality development of foreign trade and adopting the entropy method to measure the industrial policy level, Making the conclusions more robust. Therefore, the issue of appeal can be further refined in future studies.

---

## [Decision Letter · Decision Letter 1]

13 Aug 2024

Empirical Evidence on the Impact of the "New Round" of Sino-US Trade Frictions on China's Foreign Trade Industrial Policy and High-quality Development

PONE-D-24-27007R1

Dear Dr. fan,

We’re pleased to inform you that your manuscript has been judged scientifically suitable for publication and will be formally accepted for publication once it meets all outstanding technical requirements.

Kind regards,

Guangnian Xiao

Academic Editor

PLOS ONE

Additional Editor Comments (optional):

Reviewers' comments:

Reviewer's Responses to Questions

**Comments to the Author**

1. If the authors have adequately addressed your comments raised in a previous round of review and you feel that this manuscript is now acceptable for publication, you may indicate that here to bypass the “Comments to the Author” section, enter your conflict of interest statement in the “Confidential to Editor” section, and submit your "Accept" recommendation.

Reviewer #1: All comments have been addressed

Reviewer #3: All comments have been addressed

2. Is the manuscript technically sound, and do the data support the conclusions?

Reviewer #1: Partly

Reviewer #3: Yes

3. Has the statistical analysis been performed appropriately and rigorously? 

Reviewer #1: No

Reviewer #3: Yes

4. Have the authors made all data underlying the findings in their manuscript fully available?

Reviewer #1: (No Response)

Reviewer #3: Yes

5. Is the manuscript presented in an intelligible fashion and written in standard English?

Reviewer #1: Yes

Reviewer #3: Yes

6. Review Comments to the Author

Reviewer #1: comments were addressed.In that way, I do recommend to accept the manuscript to be published in the journal.

Reviewer #3: The authers have well addressed the comments from previous reviewers. Some minor typos should be fixed before publication. For instance, some sentences miss space after period （e.g. Fourth line from the bottom in Page 20; First line in Page 21）. Others like the use of "Secondly", "however" in the first paragraph in Page 21 should be checked.

7. PLOS authors have the option to publish the peer review history of their article (what does this mean?). If published, this will include your full peer review and any attached files.

Reviewer #1: No

Reviewer #3: No

---

## [Editor Report · Acceptance letter]

16 Aug 2024

PONE-D-24-27007R1 

PLOS ONE

Dear Dr. fan, 

I'm pleased to inform you that your manuscript has been deemed suitable for publication in PLOS ONE. Congratulations! Your manuscript is now being handed over to our production team.

Kind regards, 

on behalf of

Prof. Guangnian Xiao 

Academic Editor

PLOS ONE